# Targeting FLT3 Mutation in Acute Myeloid Leukemia: Current Strategies and Future Directions

**DOI:** 10.3390/cancers15082312

**Published:** 2023-04-15

**Authors:** Kateryna Fedorov, Abhishek Maiti, Marina Konopleva

**Affiliations:** 1Department of Oncology, Montefiore Medical Center, Albert Einstein College of Medicine, Bronx, NY 10467, USA; 2Department of Leukemia, The University of Texas MD Anderson Cancer Center, Houston, TX 77030, USA

**Keywords:** *FLT3*-ITD, FLT3 inhibitors, acute myeloid leukemia (AML), tyrosine kinase inhibitors (TKI)

## Abstract

**Simple Summary:**

*FLT3* mutation is commonly present in newly diagnosed patients with acute myeloid leukemia, and confers high relapse risk. Targeted therapies with FLT3 inhibitors improve survival when used as a single agent in a salvage setting, and more so when combined with other therapies in newly diagnosed patients. Development of resistance is common. New drug combinations and strategies to target FLT3 are being actively investigated.

**Abstract:**

*FLT3* mutations are present in 30% of newly diagnosed patients with acute myeloid leukemia. Two broad categories of *FLT3* mutations are ITD and TKD, with the former having substantial clinical significance. Patients with *FLT3*-ITD mutation present with a higher disease burden and have inferior overall survival, due to high relapse rates after achieving remission. The development of targeted therapies with FLT3 inhibitors over the past decade has substantially improved clinical outcomes. Currently, two FLT3 inhibitors are approved for use in patients with acute myeloid leukemia: midostaurin in the frontline setting, in combination with intensive chemotherapy; and gilteritinib as monotherapy in the relapsed refractory setting. The addition of FLT3 inhibitors to hypomethylating agents and venetoclax offers superior responses in several completed and ongoing studies, with encouraging preliminary data. However, responses to FLT3 inhibitors are of limited duration due to the emergence of resistance. A protective environment within the bone marrow makes eradication of *FLT3*^mut^ leukemic cells difficult, while prior exposure to FLT3 inhibitors leads to the development of alternative *FLT3* mutations as well as activating mutations in downstream signaling, promoting resistance to currently available therapies. Multiple novel therapeutic strategies are under investigation, including BCL-2, menin, and MERTK inhibitors, as well as FLT3-directed BiTEs and CAR-T therapy.

## 1. Introduction

FMS-related tyrosine kinase-3 (FLT3) is one of 58 human receptor tyrosine kinases. FLT3 is preferentially expressed on hematopoietic stem cells, early myeloid, and lymphoid progenitor cells [1,2]. Under normal physiologic conditions, FLT3 is activated by FLT3-ligand (FL)—a growth factor released by fibroblasts and hematopoietic cells within the bone marrow microenvironment. FL binding to FLT3 activates signaling via PI3K, STAT5, and RAS to promote cell survival, differentiation, and proliferation [3,4].

Activating mutations in *FLT3* are seen in ~30% of newly diagnosed acute myeloid leukemia (AML) patients. Two broad categories of *FLT3* mutations are: internal tandem duplication (ITD) mutations within the receptor’s autoinhibitory juxtamembrane domain (~25%); and point mutations within tyrosine kinase’s activation loop (TKD) (7–10%). While both mutations result in constitutively activated FLT3, patients with *FLT3*-ITD mutation tend to have a higher disease burden at presentation, and an inferior overall and relapse-free survival [1,3,5,6,7].

### 1.1. FLT3 Mutations in Newly Diagnosed AML

*FLT3*-ITD mutated AML is a heterogeneous entity due to variability in ITD length, insertion site, mutant-to-wild type allelic ratio (AR), overall karyotype, and co-mutations, specifically *NPM1*. Irrespective of other variables, *FLT3*-ITD mutation on its own confers a poor prognosis. A meta-analysis evaluating the prognostic significance of *FLT3*-ITD mutation prior to widespread use of FLT3 inhibitors (FLT3i) reported an overall survival hazard ratio (HR) of 1.86 and a relapse-free survival HR of 1.75 [8]. On the other hand, prognostic implications of *FLT3*-TKD mutations are not well defined, and these are generally not considered as either a favorable or an adverse feature [4]. In contrast to European LeukemiaNet (ELN) 2017 AML risk stratification criteria that classified *FLT3*-ITD mutation with AR of > 0.5 in an absence of *NPM1* mutation as an adverse risk, the updated ELN 2022 guidelines re-classify all *FLT3*-ITD mutated AML (without adverse cytogenetics) as intermediate risks, irrespective of AR or *NPM1* mutation status. The change has been implemented, given improved outcomes with the use of FLT3i and reliance on measurable residual disease (MRD) status for the therapy selection [9].

### 1.2. FLT3 Mutations in Relapsed/Refractory AML

While *FLT3* mutation is one of the most common mutations in newly diagnosed AML, it can also emerge at the time of relapse, with *FLT3*-ITD arising more commonly than *FLT3*-TKD (8% vs. 2%) [4]. It is possible that in patients with newly identified *FLT3*-ITD at the time of relapse, the mutation was present at the time of the diagnosis below the detection limit, and the *FLT3*^mut^ clone emerged as dominant under the selective stress of chemotherapy. In up to 75% of patients with *FLT3*-ITD mutation at the time of diagnosis, the mutation persists at the time of relapse, commonly with a higher allelic burden [10]. At the same time, *FLT3*-TKD mutations are more likely to disappear at the time of relapse than *FLT3*-ITD (7% vs. 4%). Similar to *FLT3*-ITD present at the time of diagnosis, the emergence of new mutation at relapse confers a poor prognosis, inferior response to salvage chemotherapy with lower rates of second complete remission (CR), and a higher rate of relapses post allogeneic stem cell transplant (allo-SCT) [4]. Currently, due to poor outcomes in the relapsed/refractory (R/R) setting, it is generally accepted that for patients with *FLT3*-ITD mutation, allo-SCT should be recommended in the first CR [2,7,9].

### 1.3. FLT3 Inhibitors

Due to the high incidence of *FLT3* mutations in AML and their association with unfavorable prognosis, multiple efforts have been directed to develop targeted therapies that would improve outcomes. The multitude of FLT3 inhibitors can be classified using two systems: first- and second-generation based on their specificity, and Type I and II inhibitors based on mechanism of their action.

The first-generation FLT3i exhibit multi-kinase target activity, which results in multiple off-target effects. Midostaurin and sorafenib are the most commonly used first-generation FLT3i in AML. The second-generation FLT3i, gilteritinib, quizartinib, and crenolanib, were designed with greater specificity, longer half-life, and potency [4]. All FLT3i prevent receptor autophosphorylation and activation of downstream signaling by interacting with the ATP-bindings site of the intracellular tyrosine kinase domain. Type 1 inhibitors, midostaurin, gilteritinib, and crenolanib, bind to both active and inactive receptor conformations, while Type 2 inhibitors sorafenib and quizartinib interact with the hydrophobic region adjacent to the ATP-binding site that is only accessible in the inactive state [4,11,12]. Type 1 inhibitors are active against both ITD and TKD mutations, while Type 2 inhibitors are only effective against ITD mutations [11]. Currently, midostaurin and gilteritinib are the only two FLT3i that have received FDA approval for the treatment of adults with *FLT3*^mut^ AML. Table 1 summarizes the characteristics of FLT3i used for the treatment of AML.

### 1.4. Synergism between FLT3i and BCL2 Inhibitors

Normal physiologic FLT3 signaling results in downstream activation of RAS/MAPK, PI3K, and STAT3 pathways promoting cell survival, proliferation, and differentiation of hematopoietic progenitor cells. It has been shown that in normal human hematopoietic stem cells, FLT3 exerts anti-apoptotic effects by maintaining high levels of MCL-1 protein. In the leukemic cell lines, *FLT3*-ITD results in MCL-1 upregulation via *FLT3*-ITD specific STAT5 activation [6].

MCL-1 upregulation has been recognized as a major resistance mechanism to antileukemic therapies targeting BCL-2. Venetoclax, a selective BCL-2 inhibitor, induces apoptosis by liberating BH3-only proteins (BIM and PUMA), allowing for BAX/BAK1 activation on the mitochondrial membrane with subsequent release of cytochrome c and activation of caspases 3 and 7 [13]. In AML, venetoclax (Ven) in combination with azacitidine yields unprecedented response rates, establishing this doublet as a new standard of care for patients not eligible for intensive chemotherapy [14]. Unfortunately, patients who progress following treatment with a combination of venetoclax and azacitidine have dismal outcomes [15]. While direct MCL-1 inhibitors are currently not available for routine use in clinical practice, indirect inhibition with FLT3i offers an attractive approach to overcoming MCL-1 mediated venetoclax resistance due to the downregulation of MCL-1 [16]. Antileukemic synergistic effects of FLT3i and venetoclax are currently being evaluated in several clinical trials.

## 2. FLT3 Inhibitors in the Front-Line Setting

### 2.1. In Combination with Intensive Chemotherapy

Since its discovery in the late 1990s, FLT3 has become widely recognized as a poor prognostic marker and an important therapeutic target. Historic data show that even though *FLT3*-ITD^mut^ patients were able to achieve CR rates over 70% with first induction, the majority of patients ultimately relapsed, with a 5-year progression free survival (PFS) rate of 20% and a 5-year overall survival (OS) rate of 14% [17]. Details of relevant clinical trials evaluating FLT3i in combination with intensive chemotherapy, are summarized in Table 2.

Midostaurin is a first-generation, Type 1 FLT3i. Midostaurin in combination with intensive chemotherapy (daunorubicin and cytarabine, “7 + 3”) was the first to be approved by the FDA in 2017 as a frontline therapy based on results of RATIFY, a phase 3 randomized, double-blind, placebo-controlled clinical trial. The study allowed patients with both *FLT3*-ITD and *FLT3*-TKD mutations, regardless of allele ratio. Therapy with midostaurin resulted in a significantly prolonged OS of 74.7 months vs.25.6 months (*p* = 0.009) in the study and placebo groups respectively. Notably, CR rates were not significantly different, at 58.9% and 53.5% for the midostaurin and placebo groups, respectively. Patients treated with midostaurin experienced higher rates of anemia and rash compared to those in the placebo arm [18].

Sorafenib is a first-generation, Type 2 FLT3i that was evaluated in combination with 7 + 3 in a phase 2 randomized, double-blind, placebo-controlled clinical trial SORAML. The study included AML patients with or without *FLT3* mutation (34% were *FLT3*
^mut^). In addition to a combination of sorafenib with 7 + 3 induction and 3 cycles of high-dose cytarabine consolidation, patients in the study group received 12 months of sorafenib maintenance. The addition of sorafenib to the 7 + 3 regimen significantly improved both event-free survival (EFS) (29 vs. 9 months, *p* = 0.013) and relapse-free survival (RFS) (56% vs. 38%, *p* = 0.017), but not OS. In an exploratory analysis, patients with *FLT3*-ITDmutation in both the study and the control group were found to have comparable EFS but improved OS and RFS in the sorafenib arm, albeit the difference was not statistically significant. Treatment with sorafenib resulted in substantial toxicity, with high rates of diarrhea, bleeding, and cardiac events [19]. More recently, the efficacy of sorafenib in combination with intensive chemotherapy was re-evaluated in patients with *FLT3*-ITDmutation. No statistically significant EFS or OS benefit following 7 + 3/sorafenib was observed, while patients who received allo-SCT had improved 2-years post-allo-SCT OS (78.5% vs. 54.2%). Patients with *FLT3*-ITD AR of >0.7 derived more benefit than those with AR 0.05–0.07 (HR 0.45 vs. 0.89) [20].

Gilteritinib has been studied in combination with anthracycline and cytarabine induction, followed by cytarabine consolidation in an open-label phase 1 study. The study enrolled patients with newly diagnosed AML, irrespective of *FLT3* mutation status. A total of 38 *FLT3*^mut^ patients in part 2 of the study received gilteritinib and achieved a composite CR (CR, CRi, CRp) rate of 81.6%, with 70% clearing *FLT3*-ITDmutation. Notably, in parts 1 and 2 of this study, patients received idarubicin in place of daunorubicin, and subsequently, in part 3, received conventional 7 + 3, with no significant difference in the outcomes. The median OS for the entire study cohort was 35.8 months. In addition, 10 of 79 (12.7%) patients had treatment-related adverse events (AE) leading to gilteritinib discontinuation, the most common grade ≥3 AE being transaminitis (13.9%), pneumonia, bacteremia, and sepsis (13.9%, 11.4%, and 11.4%, respectively) [21]. Currently, a phase 3 multicenter open-label randomized study (NCT04027309) comparing gilteritinib vs. midostaurin in combination with 7 + 3, initiated in 2019 by HOVON and AMLSG cooperative study groups, continues enrollment [22].

QuANTUM-First is a phase 3 randomized, double-blind, placebo-controlled clinical trial evaluating the efficacy of quizartinib, a 2nd-generation Type 2 FLT3i, in combination with 7 + 3 induction therapy followed by 3 years of maintenance. Recently presented preliminary data revealed significantly prolonged OS in the quizartinib arm (31.9 vs. 15.1 months, *p* = 0.0324). Patients treated with quizartinib experienced increased rates of neutropenia, infections, and QT prolongation [23], yet the safety was acceptable. Subgroup analysis of patients who had allo-SCT in the first CR (CR1) followed by maintenance, demonstrated substantial benefit in terms of OS (HR 0.326) [24].

The addition of FLTi to the intensive chemotherapy backbone of 7 + 3, is currently the standard of care for patients with *FLT3*^mut^ AML eligible for intensive chemotherapy. The addition of venetoclax to intensive chemotherapy (IC), either 7 + 3, FLAG-IDA (fludarabine, idarubicin, cytarabine), or CLIA/FIA (fludarabine, idarubicin, cytarabine), has recently been investigated in multiple clinical trials, and resulted in manageable toxicity profiles and promising outcomes [25,26]. The addition of FLT3i to the backbone of CLIA/venetoclax has also been investigated, but resulted in prohibitively prolonged myelosuppression [27]. Clinical trials evaluating the efficacy and tolerability of other induction regiments in combination with FLT3i are ongoing: CLIA (cladribine, idarubicin, cytarabine) with gilteritinib (NCT02115295), and CPX-351 with quizartinib (NCT04128748).

### 2.2. FLT3i Combination Therapies for Patients Ineligible for Intensive Chemotherapy

In the past, HMA alone have been widely utilized for the management of older patients unfit for intensive chemotherapy, albeit without substantial survival advantage when compared to the best supportive care. Older unfit patients with FLT3 mutation were treated with a combination of FLT3i and HMA (Table 2) [28,29].

Strati et al. evaluated the efficacy of midostaurin in combination with azacitidine in both newly diagnosed patients ineligible for intensive chemotherapy, and those with R/R AML. A total of 74% of patients had *FLT3* mutation, with 24% of patients having prior exposure to FLT3i (either sorafenib or quizartinib). Patients without prior exposure to FLT3i had a response rate of 33%. The presence of *FLT3* mutation did not correlate with response duration, although those without prior FLT3i exposure had longer responses (31 vs. 16 weeks) [30].

A study evaluating a combination of sorafenib and azacitidine, conducted by Ohanian et al., included elderly patients with *FLT3*-ITDAML, 44% with secondary AML. The overall response rate (CR/CRi/CRp/PR) was 78%, with a median CR duration of 14.5 months. Despite high initial response rates, the median OS was 8.3 months [31].

LACEWING was a phase 3 open-label randomized clinical trial evaluating the efficacy of gilteritinib in combination with azacitidine in *FLT3*^mut^ AML. Similar to other FLT3i/HMA doublets, gilteritinib/azacitidine did not offer a significant OS advantage. While CR rates were similar between the groups (16.2% vs. 14.3%), the composite CR rate was higher in the gilteritinib/azacitidine group (58.1% vs. 26.5%), as well as in patients with *FLT3* AR of >0.5. Notably, the primary endpoint has been confounded by salvage therapies received by patients in the azacitidine group, which included gilteritinib and other FLT3i, and by the inferior performance status of patients in the gilteritinib/azacitidine cohort [32].

Low-dose cytarabine (LDAC) and venetoclax as first-line therapy in patients ineligible for induction chemotherapy, evaluated by Wei et al., offered OS advantage compared to single agent LDAC, at 6 months follow-up [33]. More recently, the same group presented interim results of a novel “sequential” triplet regimen of LDAC/venetoclax with midostaurin. Patients were treated with low-dose cytarabine on days 1–10, followed by midostaurin on days 11–28, while receiving venetoclax continuously on days 1–28. Study enrollment was not limited to *FLT3*^mut^ patients. The median OS at 18 months of follow-up was not reached. The overall response rate (ORR) by intention to treat was 77.8% and 44.4% CR, with a median RFS of 11.7 months. The regimen was well tolerated, as evidenced by inter-cycle times of 28–35 days. A randomized phase 2 trial comparing LDAC/venetoclax/midostaurin to LDAC/venetoclax, is currently ongoing [34].

### 2.3. Maintenance after Allo-SCT

Indication for allo-SCT in *FLT3*-ITD^mut^ AML is contingent on multiple factors, such as allelic burden, concurrent mutations such as *NPM1*, MRD status, and prior use of FLT3i, as well as unique variables pertaining to the recipient, donor availability, and graft specifics. In general, due to the poor prognosis associated with *FLT*-ITD mutation in AML, the current practice recommendation is to offer allo-SCT to fit patients in CR1. Even after allo-SCT in CR1, patients with *FLT3*^mut^ AML are at high risk for relapse (30–59%) [2]. Since therapy of relapsed *FLT3*^mut^ AML is rarely effective long term, prevention of relapse with targeted maintenance therapy was investigated.

Sorafenib and midostaurin have both been evaluated in phase II trials as monotherapy maintenance post-allo-SCT in patients with *FLT3*^mut^ AML (Table 2).

SORMAIN was a phase 2 randomized placebo-controlled double-blind maintenance trial comparing single-agent sorafenib to placebo in patients with *FLT3*^mut^ AML (with or without *NPM1* mutation) who were in complete hematologic remission after undergoing allo-SCT. Sorafenib required dose escalation to 400mg twice a day over 6 weeks, and was administered continuously for a total of 24 months. Sorafenib offered a substantial OS and disease-free survival, with an HR of 0.39. While MRD-negative patients derived the most benefit (*p* = 0.028), those with MRD-positive disease also had significantly improved RFS when treated with sorafenib (*p* = 0.015) [35].

Such improved outcomes with the use of sorafenib following allo-SCT are attributed to mechanisms other than *FLT3*-ITDinhibition. A retrospective study published prior to SORAML suggests a unique synergism between sorafenib and the alloimmunity [36,37]. Owing to its multikinase activity, sorafenib downregulates activating transcription factor 4 (ATF4), which increases IL-15 production by *FLT3*-ITD^mut^ leukemic cells. IL-15 produced by leukemic cells promotes the expansion of donor-derived CD8+/CD107a+/IFN-γ + cytotoxic T cells that augment graft versus leukemia effect, possibly allowing improved and durable outcomes [38]. Gilteritinib has been shown to exert similar effects [39]. It has also been shown that increased levels of IL-15 significantly decrease PD-1 expression by T-cells and compromise self-tolerance following allo-SCT, potentially increasing the risk of graft versus host disease (GVHD). Notably, GVHD was the most common reason for treatment discontinuation in SORMAIN [35].

The results of RADIUS, a randomized phase 2 placebo-controlled open-label trial evaluating the efficacy of midostaurin maintenance in a post-allo-SCT setting, were published shortly after SORMAIN. A 12-month course of midostaurin maintenance did not result in statistically significant improvement in event-free or overall survival, with EFS of 89% vs. 76% and OS of 85% vs. 76% in study and standard of care groups, respectively. Notably, the study was not powered to detect statistical significance between the two arms [40].

Studies evaluating gilteritinib (MORPHO, NCT02997202) and crenolanib (NCT03258931) maintenance are currently ongoing.

## 3. FLT3 Inhibitors in the Relapsed Refractory Setting

While incorporation of FLT3i into upfront treatment regimens became an established practice, the efficacy of FLT3i in the relapsed refractory setting has been initially investigated as monotherapy (Table 2).

Midostaurin monotherapy in the R/R setting did not offer a survival advantage [41]. In 2019, 2 phase 3 clinical trials were published comparing quizartinib and gilteritinib to salvage chemotherapy of physician’s choice in patients with R/R *FLT3*^mut^ AML. The use of quizartinib in the R/R setting offered an OS advantage of 6.2 vs. 4.7 months for those treated with salvage chemotherapy (HR 0.76). Nevertheless, despite 48% of patients achieving CRc with quizartinib, the remissions were short-lived, with a median duration of CRc of 12.1 weeks. A total of 32% of patients treated with quizartinib subsequently underwent allo-SCT, compared to 11% in the chemotherapy group [42]. Gilteritinib monotherapy for R/R *FLT3*^mut^ AML offered a median OS of 9.3 months, compared to 5.6 months with salvage chemotherapy (HR 0.63). A total of 34.0% of patients achieved CRc with a median duration of remission of 11 months. Also, 25% of patients receiving gilteritinib underwent allo-SCT. Thus, despite apparent higher response and allo-SCT rates with quizartinib, gilteritinib monotherapy results in more durable remissions that translate into improved overall survival [43]. In 2018, gilteritinib was approved as a monotherapy for patients with R/R *FLT3*^mut^ AML.

The synergism between FLT3i and venetoclax described above has been studied in a phase 1b study in which Daver et al. evaluated the efficacy of venetoclax/gilteritinib doublet in patients with R/R *FLT3*^mut^ AML. The regimen resulted in 89% mCRc, with 60% of patients achieving molecular response with *FLT3*-ITDVAF of <10^−2^. Among patients treated, 97% experienced grade ≥3 adverse events, most commonly (>25%) cytopenias, necessitating dose interruptions or discontinuations in 80% of patients. This study offers important insight into the toxicity profile of combining venetoclax with FLT3i, with the main concern of additive myelosuppression [44].

## 4. FLT3 in Combination with HMA/Ven

Currently, the HMA/Ven doublet is an established standard of care for newly diagnosed patients with FLT3 unfit for intensive chemotherapy, but relapses inevitably occur despite high initial CR rates (comparable between *FLT3*^mut^ and *FLT3*^wt^), with *FLT3*-ITD AML having shorter remissions and lower overall survival rates [45,46]. Retrospective analysis of elderly patients enrolled in a clinical trial evaluating the addition of FLT3i to low-intensity therapy, found that the combination of FLT3i with HMA/Ven resulted in significantly improved outcomes (CR/CRi rates, MRD negativity, and OS rate), compared to HMA or LDAC regiments without venetoclax [47]. Subsequently, several prospective trials evaluated the efficacy and safety of adding FLTi to HMA/Ven, in both upfront and salvage settings (Table 2).

Maiti et al. reported on outcomes of adding FLTi of clinician’s choice to decitabine/venetoclax backbone for treatment of both newly diagnosed and R/R *FLT3*^mut^ AML. In newly diagnosed patients, the composite CR rate was 92%, including 91% MRD negativity by polymerase chain reaction/next-generation sequencing (PCR/NGS) in responders, with 80% 2-year OS. In those with R/R AML, the CRc rate was 62% with 100% MRD negativity by PCR/NGS in responders, yet a short median OS of 6.8 months. CR rates were high (63%), even in those treated with FLT3i in prior lines of therapy. Delays in count recovery and rates of neutropenic complications were comparable to those reported in the VIALE-A study.

More recently, Yilmaz et al. presented preliminary outcomes data on a decitabine, venetoclax, and quizartinib triplet in *FLT3-ITD*^mut^ AML. A total of 78% of patients with R/R AML achieved CR/CRi with a median OS of 7.6 months at a 13-month follow-up [48]. Short et al. presented data on the use of azacitidine and venetoclax in combination with gilteritinib. A total of 95% of newly diagnosed patients achieved CR with a 1-year survival rate of 80%. The CR/CRi rate in the R/R cohort was 37%, with 50% of responders clearing *FLT3* by PCR. The 1-year OS for R/R patients was 27% [49]. Both the gilteritinib, azacitidine, venetoclax, and the quizartinib, decitabine, venetoclax triplet combinations offered outstanding CR rates of >95% in patients with newly diagnosed *FLT3*^mut^ AML. Updated results of these ongoing studies are eagerly awaited.

In an effort to further improve the quality of life of patients receiving therapy for AML, an entirely oral formulation of HMA has been developed. Use of ASTX727, an oral decitabine with cedazuridine (cytidine deaminase inhibitor), in combination with venetoclax, results in an ORR of 61% in newly diagnosed patients and an ORR of 45% in patients with R/R AML [50]. Results of treatment of *FLT3*^mut^ patients with a new triplet combination of ASTX727, venetoclax, and gilteritinib, have been presented. While the sample size was small, 50% (8/8) achieved CR/CRi with an estimated 6-month OS of 70%. Novel regimens had no dose-limiting toxicities but did result in significant myelosuppression, with the majority of grade 3 events due to infectious complications [51].

## 5. Measurable Residual Disease in *FLT3*^mut^ AML

Measurable residual disease is an important prognostic and predictive biomarker in AML. Currently, multiparametric flow cytometry (MFC) and polymerase chain reaction (PCR) are the most commonly used MRD assays. MFC relies on the identification of diagnostic leukemia associated immune phenotype (LAIP) and different from normal (DfN) immunophenotype, with sensitivity ranging from 1 × 10^−4^ to 1 × 10^−5^. The use of PCR for MRD detection in AML offers more sensitivity (1 × 10^−6^) compared to MFC but is limited by the presence of targetable genetic abnormalities, and is currently primarily used in *NPM1* mutated AML [9,52]. Next generation sequencing (NGS) is an emerging and promising MRD assessment technique increasingly used in clinical trials, but at this time, it has limited sensitivity, which precludes its expansion into clinical practice. In *FLT3*^mut^ AML, MRD status is highly predictive of outcomes and thus is used to guide the therapy [53]. Unfortunately, at this time, PCR and NGS are lacking sensitivity for *FLT3*-ITDMRD detection due to the heterogeneity of ITDs.

A new PCR-NGS MRD assay for *FLT3*-ITDdetection was evaluated in patients enrolled in a RATIFY trial. The new assay was able to detect all *FLT3*-ITDmutations that were also identified by conventional capillary electrophoresis (CE) PCR. Additionally, it identified a patient in remission with persistent *FLT3* mutation that was not identified by CE PCR [54]. Subsequently, this PCR-NGS assay was used to evaluate the clinical outcomes of QuANTUM-First trial. An MRD analysis was performed on samples from 317 out of 368 patients that achieved CR after 1–2 courses of induction. CRc with an MRD of <10^−4^ correlated with improved overall survival. While the proportion of patients with an MRD of <10^−4^ was comparable between study and control groups (24.6% vs. 21.4%, *p* = 0.385), the proportion of patients with an undetectable MRD of <10^−5^ using PCR-NGS technique was significantly higher following treatment with quizartinib (13.8% vs. 7.4%, *p* = 0.017), indicating the improved depth of remission with the use of FLT3i [55].

## 6. Mechanisms of Resistance

Remarkable progress has been made in developing a multitude of FLT3i that offer improved relapse-free and overall survival in patients with this high-risk AML. Unfortunately, resistance to FLTi is common, and can be classified as either primary or secondary. Primary resistance (or innate resistance) is attributed to mutations present prior to initiation of FLTi, characterizing such patients as non-responders. Secondary resistance is acquired after exposure to FLT3i and is commonly seen following a relapse [12,56].

### 6.1. Primary Resistance Mechanisms

Primary resistance to FLT3i is multifactorial, with molecular signaling, bone marrow stromal microenvironment, and drug metabolism being the key mechanisms [57]. The bone marrow microenvironment serves as a protective niche for *FLT3*^mut^ AML cells, as suggested by the rapid eradication of leukemic blasts from the periphery but not from the bone marrow. Stroma-derived cytokines and/or direct contact with stromal cells allows for FLT3-ITDindependent activation of the RAS/MEK/ERK pathway, contributing to the survival of leukemic blasts within the bone marrow despite FLT3 inhibition [58]. FLT3-ligand, a growth factor produced by bone marrow stromal cells and T-lymphocytes, plays an important role in the homeostasis of hematopoietic progenitor cells [59]. FL rises when the bone marrow niche is compromised, and its levels are inversely proportional to the fraction of *FLT3*^mut^ blasts. Thus, FL levels are low pre-treatment and increase during the aplastic phase following cytotoxic chemotherapy [60,61]. Such an increase in FL levels following chemotherapy has been recognized as one of the most common mechanisms of primary resistance. Wild type (WT)-FLT3 is commonly co-expressed alongside mutated *FLT3* on leukemic cells. In the post-induction phase, binding of abundant FL to WT-FLT3 allows for downstream signaling via activation of an RAS/MEK/ERK pathway nourishing and promoting proliferation of remailing *FLT3*^mut^ leukemic cells despite FLT3i (Figure 1, pathway 1) [59,62].

Fibroblast growth factor-2 (FGF-2) and CCR4 have also been implicated in FLT3i resistance. FGF-2 secreted from bone marrow stromal cells activates FGFR-1, resulting in activation of MAPK signaling, allowing for leukemic cell proliferation (Figure 1, pathway 2) [57]. Additionally, CXCR4 expressed on leukemic cells, and its ligand CXCR12, facilitate their homing to the bone marrow where they are nourished within the environment rich in growth-promoting and anti-apoptotic signals (Figure 1, pathway 3) [63].

FLTi are primarily metabolized by hepatic CYP3A4, and their bioavailability may vary with concurrent use of moderate and strong CYP3A4 inhibitors [64,65]. Bone marrow stromal cells have been found to express CYP3A4, where they protect hematopoietic cells from toxic insults. Thus, local inactivation of FLT3i by CYP3A4 within the bone marrow environment, is yet another mechanism of resistance (Figure 1, pathway 4) [66].

### 6.2. Secondary Resistance Mechanisms

Secondary resistance to FLT3i can be classified as on-target, referring to changes in FLT3 itself (23%), and off-target mutations in oncogenic pathways not directly dependent on FLT3, such as epigenetic modifiers (16%), RAS/MAPK pathway genes (13%), TP53(7%), and WT1 (7%) [12].

On-target resistance to FLT3i is characterized by the acquisition of *FLT3* mutations involving the activating loop or the gatekeeper residues [12]. There are three mechanisms via which such mutation results in drug resistance. The first mechanism leads to decreased drug binding due to changes in the FLT3 residues that directly interact with FLTi, with the “gatekeeper” F691 mutation being a classic example that confers resistance to all FLT3 inhibitors currently in clinical use. The second mechanism is the stabilization of active conformation of FLT3 caused by mutations Y842H and D835F resulting in resistance to both quizartinib and sorafenib. The third mechanism makes the FLT3 more active, increasing the speed of downstream reactions and/or increased affinity to its substrate. An example of such activating mutation is M664I, which confers resistance to pexidartinib—Type 2 inhibitor that has activity against *FLT3* with F691 mutation (Figure 1, pathway 5) [1,57,59].

Off-target resistance to FLT3i can be mostly attributed to activating mutations in the RAS/MAPK signaling pathway, specifically mutations in *NRAS, KRAS, PTPN11, CBL*, and *BRAF* (Figure 1, pathway 6). Off-target mutations in the RAS/MAPK pathway and on-target mutations in *FLT3*, with the exception of *FLT-F691L*, are not mutually exclusive. Up to 12% of patients resistant to gilteritinib no longer express mutated FLT3, yet harbor mutations in the RAS/MAPK pathway, while up to 27% of patients have persistence of *FLT3* mutation as well as new emergent RAS/MAPK mutations [57]. Autophagy has been recognized as one of the survival mechanisms utilized by *FLT3*-ITD AML cells and an important resistance mechanism to FLT3 inhibition. Autophagy is induced under hypoxic conditions and bone marrow microenvironment via BTK signaling [67]. Mutations in D835Y acquired after treatment with sorafenib have been associated with increased autophagy markers [68].

## 7. Real-Life Experience with FLT3i Resistance and Clinical Implications

Alotaibi et al. compared mutational profiles of patients with AML before and after FLT3i-based therapy (midostaurin, gilteritinib, crenolanib, quizartinib, and sorafenib). They found that exposure to different types of FLT3i (i.e., Type 1 or Type 2), in combination with either cytotoxic chemotherapy or low-intensity therapy, leads to different resistance mechanisms [12].

Treatment with Type 2 FLT3i most commonly results in the acquisition of *FLT3-D835* (30%), *IDH1/2* (10%), and *TP53* (10%) mutations, with different patterns observed depending on the backbone to which FLT3i was added. Emergent *FLT3-D835* mutation is associated with decreased overall survival of only 2.6 months, compared to 6.7 for relapsed patients without *FLT3-D835* mutation (*p* = 0.002). Patients treated with a combination of cytotoxic chemotherapy and Type 2 FLTi were more prone to the development of *TP53* (18%), *WT1* (18%), and *DNMT3A* (12%) mutations. On the other hand, *FLT3-D835* (45%) and *IDH1/2* (17%) were more common in those treated with a combination of less intensive therapy and Type 2 FLT3i.

The most common emergent mutations following treatment with Type 1 FLT3i were in the RAS/MAPK pathway and were more commonly seen when combined with low-intensity chemotherapy (29%). Relapsed patients with *RAS/MAPK* mutations had an inferior overall survival rate of 2.4 months compared to those without, who had an overall survival rate of 6.8 months (*p* = 0.009).

Up to 26% of patients no longer express FLT3 at relapse following FLT3i-based therapy. This was seen regardless of the type of FLT3i or the chemotherapy backbone used. Those with undetectable *FLT3* mutation at relapse had a better overall survival rate of 4.6 vs. 9.9 months (*p* = 0.029).

## 8. Future Directions

As mentioned previously, the prognostic implications of *FLT3* mutation in AML depend on cytogenetics and co-occurrence with *NPM1* mutation. Gene expression profiles of *NPM1*^mut^ and *MLL*-rearranged AML reveal overlapping expression of leukemogenic genes, including *MEIS1*, *HOX*, *PBX6*, and *FLT3* [69]. While there are currently no approved targeted therapies for *NPM1* mutated AML, over the past couple of years, menin inhibitors have emerged as a promising therapy for patients with *NPM1*^mut^ or *MLL*-rearranged AML. Given the synergistic leukemogenic effects of *FLT3* and *NPM1* mutations, dual targeting with FLT3i and menin inhibitors is being investigated. Pre-clinical studies demonstrate strong synergistic effects of the menin-FLT3 inhibitor doublet, with significant reduction of leukemic burden, suppression of downstream genes, and long-lasting responses [70,71]. A combination of menin inhibitors with venetoclax has also been shown to have strong antileukemic activity in mice xenografted with patient-derived *NPM1*^mut^/*FLT3*^mut^ leukemic cells, albeit prolonged exposure to menin inhibitors and venetoclax increased levels of phosphorylated FLT3, contributing to resistance. Further research is needed to evaluate the efficacy of triplet therapy with menin inhibitors, FLT3i, and venetoclax [72].

Receptor tyrosine kinase anexelecto (AXL) is critical for FLT3 signaling. A dual AXL and MER Tyrosine Kinase (MERTK) inhibitor ONO-7475 has strong antileukemic effects in *FLT3*-ITD AML cells, by suppressing ERK1/2 phosphorylation and MCL-1 expression. The combination of ONO-7475 with venetoclax overcomes venetoclax resistance, decreasing levels of pro-growth and pro-apoptotic proteins [73]. The success of ONO-7475 in *FLT3*^mut^ AML resistant to venetoclax is promising, but its effectiveness in the clinical setting is yet to be investigated.

As the development of resistance to FLT3i is common, immense effort is directed to evaluate the pre-clinical efficacy of various treatment combinations in AML resistant to currently approved FLT3 therapies. One such novel combination employs BTKi and FLT3i. BTK inhibition with ibrutinib appears to enhance quizartinib-mediated apoptosis even in *FLT3*-ITD/D835Y mutated AML cells [30,59]. Aurora kinase inhibition is also being actively investigated for the treatment of *FLT3*^mut^ AML. Dual targeting of FLT3 and aurora kinase inhibits the proliferation of both wild-type and mutated FLT3 AML cell lines. Moreover, such dual kinase inhibition overcomes selective FLT3i resistance in cell lines with D835Y mutations, and maintains its efficacy even after prolonged exposure [74].

Bispecific T-cell engager (BiTE) and chimeric antibody receptor T-cell (CAR-T) therapy have become commonplace for patients with hematologic malignancies, except AML, where lack of suitable surface target unique to AML cells has been a major limitation. FLT3 is expressed on virtually all AML cells, making it an attractive target for both BiTE and CAR-T therapy. Unfortunately, FLT3 expression is not limited to AML cells, as it is also expressed on hematopoietic stem cells as well as lymphoid progenitor cells, limiting target specificity and increasing risks of substantial myelosuppression. Additionally, targeting only *FLT3*-ITD/TKD mutated cells is not possible, as the altered portion of the protein is intracellular and not available for interaction with BiTE or CAR-T.

Nevertheless, both FLT3 BiTE and CAR-T are currently being developed and tested in both pre-clinical and clinical settings. AMG-427 is a FLT3 BiTE that showed promising anti-leukemic activity in pre-clinical studies. Its activity was ultimately limited by the upregulation of PD-1 by AML cells, but was effectively counteracted by PD-1 blockade [75]. AMG-427 is being evaluated in a phase I clinical trial (NCT03541369) in adults with relapsed refractory AML. At least seven FLT3-directed CAR-T are under development. The majority of FLT3 CAR-Ts are second-generation. They employ either CD28 or 4-1BB co-stimulatory domains and vary broadly in their single-chain variable fragment (scFv). Only three FLT3 CAR-Ts are ready for testing in a clinical setting [76]. Two clinical trials are currently recruiting in China. The clinical trial of AMG-533 (NCT03904069) is anticipated to start recruiting in the US in early March 2023.

## 9. Conclusions

FLT3 mutated AML is a common, physiologically complex, and clinically aggressive disease. A detailed understanding of cellular signaling pathways, microenvironment interactions, and the role of the immune system is needed for the development of effective therapies for *FLT3*^mut^ AML. Multitudes of FLT3 inhibitors have already entered clinical practice and are being combined with other approved therapies in an effort to maximize the benefits. While initial remission rates with current therapies are high, resistance to FLT3i is a major limitation to long-term survival. The development of therapies offering durable responses is of paramount importance.

## Figures and Tables

**Figure 1 cancers-15-02312-f001:**
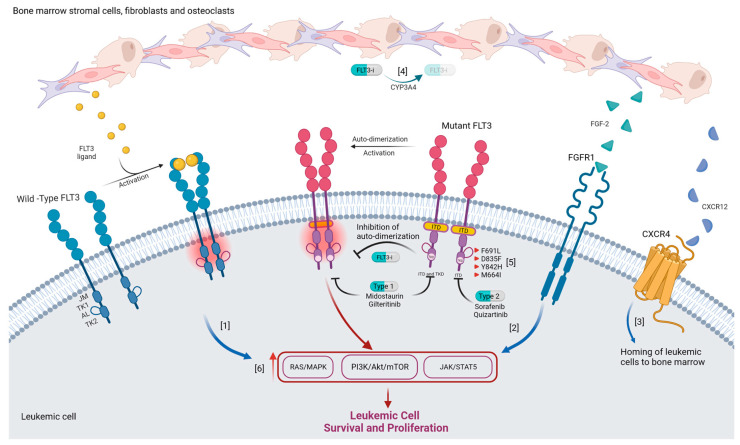
Mechanisms of resistance to FLT3i. 1. Increase in FLT3 ligand levels following induction chemotherapy results in increased RAS/MEK/ERK signaling following its activation of wild-type FLT3 receptor. 2. FGF-2 secreted from bone marrow stromal cells activates FGFR-1 resulting in the activation of MAPK signaling. 3. Binding of CXCR12 released by osteoblasts to CXCR4 expressed on leukemic cells promotes their homing to bone marrow. 4. Increased metabolism of FLT3i by CYP34A decreases drug efficacy. 5. On target mutations promote resistance to FLT3i: F691L mutation causes decreased drug binding, D835F and Y842H stabilize active conformation of FLT3, and M664I increases speed of downstream signaling. 6. Activating mutation in components of downstream signaling pathways contributes to cell survival and proliferation independent of FLT3 signaling.

**Table 1 cancers-15-02312-t001:** Characteristics of FLT3 inhibitors.

Drug	Type	Generation	Receptor Sensitivity	Mutation Sensitivity to FLT3i		
				ITD	D835Y	F691L
Midostaurin	1	First	Low	S	S	R
Gilteritinib	1	Second	Moderate	S	S	R
Crenolanib	1	Second	Moderate	S	S	R
Sorafenib	2	First	Moderate	S	R	R
Quizartinib	2	Second	High	S	R	R

**Table 2 cancers-15-02312-t002:** Summary of relevant clinical trials using FLT3 inhibitors in AML.

	Regimen	Study Name NCT#	Mutation	Author, Year	Phase	Blinding	n	Response	MRD Negativity	Survival
Eligible for IC Frontline Not eligible for IC	7 + 3 + Midostaurin	RATIFY	ITD/TKD	Stone, 2017	3	Double blind	360 vs 357	CR: 58.9% vs 53.5%	-	OS: 74.7 vs. 25.6 mo
	7 + 3 + Sorafenib	SORAML	All, 34% *FLT3*^mut^	Rollig, 2015	2	Double blind	134 vs 133	-	-	EFS: 21 vs. 9 mo
	7 + 3 + Gilteritinib	NCT02236013	All	Pratz, 2020	1	Open label	80	CR/CRi/CRp: 81.6%	-	OS: 35.8 mo
	7 + 3 + Quizartinib	QuANTUM-First	ITD	Erba, ongoing	3	Double blind	268 vs 271	CR/CRi: 71.6% vs 69.4%	PCR-NGS *: 24.6% vs 21.4%	OS: 31.9 vs. 15.1 mo
	Aza + Midostaurin	-	All, 74% *FLT3*^mut^	Strati, 2015	1/2	Open label	14/40	CR/CRi: 13%	-	OS: 22 wks
	Aza + Sorafenib	NCT02196857 and NCT01254890	ITD	Ohanian, 2017	1 + 2	-	27	CR/CRi: 70%	-	OS: 8.3 mo
	Aza + Gilteritinib	Lacewing	ITD/TKD	Wang, 2022	3	Open label	74 vs 49	CR: 16.2 vs 14.3%	-	OS: 9.8 vs. 8.9 mo
Maintenance Post-HSCT	Sorafenib	SOMAIN	ITD	Burchert, 2020	2	Double blind	43 vs 40	24 mo RFS: 85% vs 63%	-	55 mo OS: NR
	Midostaurin	RADIUS	ITD	Maziarz, 2021	2	Open label	30 vs 30	18 mo RFS: 89% vs 76%	-	24 mo OS: NR
Relapsed/Refractory	Midostaurin	NCT00045942	All, 71% *FLT3*^mut^	Fischer, 2010	2b	Open label	95	OR (FLT^mut^): 71% OR (FLT3^wt^): 56%	-	OS: 130 d
	Quizartinib	QuANTUM-R	ITD	Cortes, 2019	3	Open label	245 vs 122	HSCT bridge: 32% vs 11%	-	OS: 6.2 vs. 4.7 mo
	Gilteritinib	Admiral	ITD/TKD	Perl, 2019	3	Open label	247 vs 124	CR/CRi: 34% vs 15.3%	-	OS: 9.3 vs. 5.6 mo
Frontline and/or Relapsed/Refractory	Ven + Gilteritinib	NCT03625505	ITD/TKD	Daver, 2022	1b	Open label	61 (R/R)	CR/CRi/CRp: 40%	NGS: 42.9%	OS: 10 mo
	Dec + Ven + FLT3i	NCT03404193	ITD/TKD	Maiti, 2021	2	Open label	12 (ND) 13 (R/R)	CRc(ND): 92% CRc(R/R): 63%	MFC: 56%, PCR/NGS: 91% MFC: 63%, PCR/NGS: 100%	OS (ND): NR OS (R/R) 6.8mo
	Dec + Ven+ Quizartinib	NCT03661307	ITD	Yilmaz, ongoing	1/2	Open label	5 (ND) 23 (R/R)	CRc(ND): 100% CRc(R/R): 78%	MFC: 50%, PCR: 80% MFC: 27.8%, PCR: 37.5%	OS (ND): 14.5 mo OS (R/R): 7.6 mo
	Aza + Ven + Gilteritinib	NCT04140487	ITD/TKD	Short, ongoing	1/2	Open label	21 (ND) 19 (R/R)	CR(ND): 95% CR/CRi (R/R): 37%	MFC: 81%, PCR: 90% MFC: 43%, PCR: 50%	1 year OS (ND): 80% 1 year OS (R/R): 27%
	ASTX727 + Ven + Gilteritinib	NCT05010122	ITD/TKD	Ong, ongoing	1/2	Open label	8 (R/R)	CR/CRi: 50%	-	OS: NR

* Novel PCR-NGS technique with sensitivity of 10^−5^, MRD cut off of <10^−4^; Aza = azacitidine, CR = complete remission, CRi = complete remission with incomplete count recovery, CRp = complete remission with partial count recovery, Dec = decitabine, FLT3i = FLT3 inhibitor, IC = intensive chemotherapy, MRD = measurable residual disease, ND = newly diagnosed, NGS = next generation sequencing, NR = not reached, OR = overall response, OS = overall survival, PCR = polymerase chain reaction, Ven = venetoclax.

## Data Availability

Not applicable.

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
