# Peer review of "Targeting FLT3 Mutation in Acute Myeloid Leukemia: Current Strategies and Future Directions"

_cancers, 2023, doi:10.3390/cancers15082312_

Round 1
Reviewer 1 Report
This review provides a comprehensive overview of the current understanding of FLT3 mutations and the development of targeted therapies with FLT3 inhibitors. As the authors mentioned, synergistic leukemogenic effects of FLT3 and NPM1 mutations. Combination of FLT3i and menin inhibitors is a promising therapy for patients with Flt2 and NPM1mut.
The authors summarized many published reports. while the article lacks the author's own thinking and opinions. The review also highlights the challenges associated with FLT3 inhibitors, particularly with regard to the chemotherapy resistance. The discussion of mechanisms of resistance is similar to the previous reviews and does not propose a new perspective.
This review article is not well structured. Some issues are listed below.
1.It’s better for the authors to plot figures to illustrate the Flt3-ITD and TKD mutations and mechanisms of treating AML with first generation and second generation FLT3i.
2.For the FLT3i combination therapies with intensive chemotherapy and for patients ineligible for intensive chemotherapy, the authors should make a table for these FLT3i therapies with information of CR, OS, MRD. Long paragraphs of text are tedious to the reader.
Author Response
Dear Reviewer #1
Thank you for taking your time to review our manuscript and to provide thoughtful suggestions and critique. We have revised the manuscript the accordingly. Below please find our responses to your comments.
1.It’s better for the authors to plot figures to illustrate the Flt3-ITD and TKD mutations and mechanisms of treating AML with first generation and second generation FLT3i.
Thank you for this suggestion. We generated a figure summarizing mechanism of resistance to FLTi. In the same figure we specified that Type 1 inhibitors work on both active and inactive FLT3 conformations and on both ITD and TKD mutations, while Type 2 inhibitors are only effective against inactive conformation with ITD mutation. Please see the figure within the manuscript file.
We did not include details on generations of FLT3i in the figure as they differ in target specificity rather than mechanism of action. We agree that it is important to summarize classification of different FLT3i thus we included this information in Table 1. Additionally, we moved the table to be located earlier in the text, this way it can be easily referenced by reader when they are reading about the clinical trials discussed later in the manuscript.
2.For the FLT3i combination therapies with intensive chemotherapy and for patients ineligible for intensive chemotherapy, the authors should make a table for these FLT3i therapies with information of CR, OS, MRD. Long paragraphs of text are tedious to the reader.
Thank you for this suggestion. We included data on MRD negativity in Table 2 whenever available.
Additionally, we separated some long paragraphs as per reviewer’s critique whenever possible.

Reviewer 2 Report
A useful review describing the latest updates in terms of FLT inhibition in acute myeloid leukemia. However, a few suggestions:
-I suggest deleting the abbreviation “AML” in the abstract
- In “A meta-analysis evaluating prognostic significance of FLT3-ITD muta-tion prior to widespread use of FLT3 inhibitors (FLT3i) reported overall survival HR of 1.86 and relapse free survival HR of 1.75” sentence, what about median OS and PFs?
-In “It is possible that in patients with newly identified FLT3-ITD at the time of relapse the mutation was also present at the time of the diagnosis but was below the detection limit and under selective stress of chemotherapy FLT3mut clone emerged as a dominant”, I suggest eliminating “a”
-I suggest editing “Due to commonality of FLT3 mutations in AML and poor prognosis that it denotes great effort has been made to develop targeted therapies that would alter clinical course and improve outcomes” because it’s unclear
-In “ Since its discovery in late 1990s FLT3 has become widely recognized as a poor prog-nostic marker and an important therapeutic target. Historic data show that even though FLT3-ITDmut patients were able to achieve CR rates over 70% with first induction, majority of patients ultimately relapsed with 5-year OS rates below 20% [13].”, what about PFS?
-I suggest eliminating this sentence: “Development of FLT3 inhibitors contributed significantly to therapeutic landscape of FLT3mut AML”
-Please clarify the terms of the comparison between groups in this period “Sorafenib – _first generation, type 2 FLT3i was evaluated in combination with 7+3 in phase 2 randomized, double blind, placebo-controlled, clinical trial SORALM. The study included AML patients with or without FLT3 mutation (only 34% were FLT3mut). Addition of sorafenib to 7+3 significantly improved both EFS (29 vs 9 months, p=0.013) and RFS (56% vs 38%, p=0.017), but not OS.”
-Please elaborate in more detail, adding a reference, the period :“ A phase 3 multicenter open-label randomized study (NCT04027309) comparing gilteritinib vs midostaurin in combination with 7+3 has been initiated in 2019 but enroll-ment was placed on hold in 1/2023 as per advice of data and safety monitoring board (DSMB).”
-In “ AML. 74% of” Please modify the beginning of the sentence with a number
-In “Overall response rate was 77.8% with 44.4% CR, with relapse free survival of 11.7 months.”, is it median pfs?
-Please add a reference in “ Even after allo-SCT in CR1 patients with FLT3mut AML are at high risk for relapse (30-59%). Therapy of re-lapsed FLT3mut AML is rarely effective long term, thus prevention of relapse with targeted maintenance therapy was investigated.”
- I suggest explaining better the role of sorafenib in “ SORMAIN trial was a phase 2 randomized placebo-controlled double-blind trial comparing single agent sorafenib to placebo in patients with FLT3mut AML (with or with-out NPM1 mutation) who have undergone allo-SCT and were in complete hematologic remission. Sorafenib offered a substantial overall and disease-free survival with combined HR of 0.39. MRD negativity prior to allo-SCT was associated with survival advantage in patients on sorafenib maintenance compared to placebo (p=0.028). Relapse free survival benefit was observed even patients with pre allo-SCT MRD positivity (p=0.015). Most common reason for treatment discontinuation was toxicity, with graft versus host disease (GVHD) being the most common cause [26].”, because the period is unclear
- Please clarify this period: “. Several MCL-1 inhibitors are under development but have yet to be used in day-to-day clinical practice [36]. FLT3 inhibition resulting in decrease of MCL-1 offers an attrac-tive approach to overcoming MCL-1 mediated venetoclax resistance. Antileukemic syn-ergistic effects of FLT3i and venetoclax are currently being evaluated in multiple clinical trials.”
Please clarify :” MRD analysis was performed on 317 of 368 patients that achieved CR after 1-2 courses of induction. CRc with MRD of <10-4 correlated with improved over-all survival. While proportion of patients with MRD<104 was comparable between study and control groups, the rate of MRD negativity was significantly higher following treat-ment with quizartinib (13.8% vs 7.4%, p=0.017) [51].”
-After “CXCR4 expressed on leukemic cells and its ligand CXCR12 facilitate their homing to the bone marrow where they are nourished within the environment rich in growth-promoting and anti-apoptotic signals [59].”, what about the aforementioned drug metabolism ?
- I suggest eliminating this period “Understanding the drivers for emergence of secondary resistance mechanisms are critical to overcoming them.”
-I think it might be interesting to add figures summarizing the mechanisms of resistance
Thank you
Author Response
Dear Reviewer #2
Thank you for taking your time to review our manuscript and to provide thoughtful suggestions and critique. We have revised the manuscript the accordingly. Below please find our responses to your comments.
-I suggest deleting the abbreviation “AML” in the abstract
Abbreviation “AML” has been deleted from the abstract as per your suggestion.
- In “A meta-analysis evaluating prognostic significance of FLT3-ITD mutation prior to widespread use of FLT3 inhibitors (FLT3i) reported overall survival HR of 1.86 and relapse free survival HR of 1.75” sentence, what about median OS and PFs?
We agree with the reviewer that information on median OS and PFS would be useful. We reviewed the meta-analysis article but unfortunately data on median OS and PFS time was not provided.
-In “It is possible that in patients with newly identified FLT3-ITD at the time of relapse the mutation was also present at the time of the diagnosis but was below the detection limit and under selective stress of chemotherapy FLT3mut clone emerged as a dominant”, I suggest eliminating “a”
We eliminated “a” from the sentence per reviewer’s suggestion. Currently the sentence reads as follows:
It is possible that in patients with newly identified FLT3-ITD at the time of relapse the mutation was present at the time of the diagnosis below the detection limit, and FLT3mut clone emerged as dominant under selective stress of chemotherapy.
-I suggest editing “Due to commonality of FLT3 mutations in AML and poor prognosis that it denotes great effort has been made to develop targeted therapies that would alter clinical course and improve outcomes” because it’s unclear
We have rephrased the paragraph as per your suggestion as following:
Due to high incidence of FLT3 mutation in AML and its association with unfavorable prognosis multiple efforts have been directed to develop targeted therapies that would improve the outcomes. The multitude of FLT3 inhibitors can be classified using two systems: first- and second-generation based on their specificity and type I and II inhibitors based on mechanism of action.
-In “ Since its discovery in late 1990s FLT3 has become widely recognized as a poor prognostic marker and an important therapeutic target. Historic data show that even though FLT3-ITDmut patients were able to achieve CR rates over 70% with first induction, majority of patients ultimately relapsed with 5-year OS rates below 20% [13].”, what about PFS?
Per reviewer’s suggestion added details on PFS – 20% and specified OS – 14%. Currently the paragraph is as follows:
Since its discovery in late 1990s FLT3 has become widely recognized as a poor prognostic marker and an important therapeutic target. Historic data show that even though FLT3-ITDmut patients were able to achieve CR rates over 70% with first induction, majority of patients ultimately relapsed with 5-year progression free survival (PFS) of 20% and 5-year overall survival (OS) rates of 14% [20]. Details of relevant clinical trials evaluating FLT3i in combination with intensive chemotherapy are summarized in Table 2.
-I suggest eliminating this sentence: “Development of FLT3 inhibitors contributed significantly to therapeutic landscape of FLT3mut AML”
The sentence has been eliminated. Please see the current structure of the paragraph in the response to the above comment.
-Please clarify the terms of the comparison between groups in this period “Sorafenib – _first generation, type 2 FLT3i was evaluated in combination with 7+3 in phase 2 randomized, double blind, placebo-controlled, clinical trial SORAML. The study included AML patients with or without FLT3 mutation (only 34% were FLT3mut). Addition of sorafenib to 7+3 significantly improved both EFS (29 vs 9 months, p=0.013) and RFS (56% vs 38%, p=0.017), but not OS.”
The primary analysis included both FLT3mut and wild type patients (17% in each group). Exploratory analysis was performed to investigate whether FLT3mut patients derived more benefit; it was found that EFS was comparable while RFS and OS were improved in the sorafenib arm but not statistically significant. We included this information in the text as follows:
In an exploratory analysis, patients with FLT3-ITD mutation in both study and control group were found to have comparable EFS, but improved OS and RFS in sorafenib arm, albeit the difference was not statistically significant.
-Please elaborate in more detail, adding a reference, the period :“ A phase 3 multicenter open-label randomized study (NCT04027309) comparing gilteritinib vs midostaurin in combination with 7+3 has been initiated in 2019 by HOVON and AMLSG cooperative study groups continues enrollment.”
While preparing this manuscript we found out that the enrollment has resumed. We have adjusted the text and induced the reference as follows.
Currently, a phase 3 multicenter open-label randomized study (NCT04027309) comparing gilteritinib vs midostaurin in combination with 7+3 initiated in 2019 by HOVON and AMLSG cooperative study groups continues enrollment [25].
[25] ClinicalTrials.gov. A Study of Gilteritinib Versus Midostaurin in Combination With Induction and Consolidation Therapy Followed by One-year Maintenance in Patients With Newly Diagnosed Acute Myeloid Leukemia or Myelodysplastic Syndromes With Excess Blasts-2 With FLT3 Mutations Eligible for Intensive Chemotherapy (HOVON 156 AML) 2023 [Available from: https://clinicaltrials.gov/ct2/show/NCT04027309]
-In “ AML. 74% of” Please modify the beginning of the sentence with a number
Sentence has been adjusted to not start with a number. Currently reads as follows:
Seventy-four percent of patients had FLT3 mutation, with 24% of patients having prior exposure to FLT3i (either sorafenib or quizartinib).
Additionally, we identified similar issue in other sentences and have adjusted them accordingly.
-In “Overall response rate was 77.8% with 44.4% CR, with relapse free survival of 11.7 months.”, is it median pfs?
As per reviewer’s suggestion we reviewed the cited text and adjusted the sentence as follows:
Overall response rate by intention to treat was 77.8% with 44.4% CR and median relapse free survival of 11.7 months.
-Please add a reference in “ Even after allo-SCT in CR1 patients with FLT3mut AML are at high risk for relapse (30-59%). Therapy of re-lapsed FLT3mut AML is rarely effective long term, thus prevention of relapse with targeted maintenance therapy was investigated.”
Mistakenly the reference was placed at the end of preceding sentence. We have moved it accordingly. Currently text is as follows:
In general, due to poor prognosis associated with FLT-ITD mutation in AML the current practice recommendation is to offer allo-SCT to fit patients in CR1. Even after allo-SCT in CR1 patients with FLT3mut AML are at high risk for relapse (30-59%) [2]. Since, therapy of relapsed FLT3mut AML is rarely effective long term, prevention of relapse with targeted maintenance therapy was investigated.
- I suggest explaining better the role of sorafenib in “ SORMAIN trial was a phase 2 randomized placebo-controlled double-blind trial comparing single agent sorafenib to placebo in patients with FLT3mut AML (with or with-out NPM1 mutation) who have undergone allo-SCT and were in complete hematologic remission. Sorafenib offered a substantial overall and disease-free survival with combined HR of 0.39. MRD negativity prior to allo-SCT was associated with survival advantage in patients on sorafenib maintenance compared to placebo (p=0.028). Relapse free survival benefit was observed even patients with pre allo-SCT MRD positivity (p=0.015). Most common reason for treatment discontinuation was toxicity, with graft versus host disease (GVHD) being the most common cause [26].”, because the period is unclear
As per reviewer’s recommendation we have revised the paragraph and included information about synergism of sorafenib and allo-immunity; We hope this better explains the role of sorafenib in the trial.
SORMAIN was a phase 2 randomized placebo-controlled double-blind maintenance trial comparing single agent sorafenib to placebo in patients with FLT3mut AML (with or without NPM1 mutation) who were in complete hematologic remission after undergoing allo-SCT. Sorafenib required dose escalation to 400mg twice a day over 6 weeks and was administered continuously for total of 24 moths. Sorafenib offered a substantial OS and disease-free survival with HR of 0.39. While MRD negative patients derived most benefit (p=0.028), those with MRD positive disease also had significantly improved RFS when treated with sorafenib (p=0.015)[37].
Such improved outcomes with use of sorafenib following allo-SCT are attributed to mechanisms other than FLT3-ITD inhibition. Retrospective study published prior to SORAML suggest unique synergism between sorafenib and allo-immunity [38, 39]. Owing to its multikinase activity sorafenib downregulates activating transcription factor 4 (ATF4) which increases IL-15 production by FLT3-ITDmut leukemic cells. IL-15 produced by leukemic cells promotes expansion of donor-derived CD8+/CD107a+/IFN-γ+ cytotoxic T cells that augment graft versus leukemia effect possibly allowing improved and durable outcomes [40]. Gilteritinib has been shown to exert similar effects [41]. It has also been shown that increased levels of IL-15 significantly decrease PD-1 expression by T-cells and compromise self-tolerance following allo-SCT potentially increasing the risk of graft versus host disease (GVHD). Notably, GVHD was the most common reason for treatment discontinuation in SORMAIN [37].
- Please clarify this period: “. Several MCL-1 inhibitors are under development but have yet to be used in day-to-day clinical practice [36]. FLT3 inhibition resulting in decrease of MCL-1 offers an attractive approach to overcoming MCL-1 mediated venetoclax resistance. Antileukemic synergistic effects of FLT3i and venetoclax are currently being evaluated in multiple clinical trials.”
Paragraph has been rephrased to as per reviewer’s suggestion. It currently reads as follows:
While direct MCL-1 inhibitors are currently not available for routine use in clinical practice, indirect inhibition with FLT3i offers an attractive approach to overcoming MCL-1 mediated venetoclax resistance, due to downregulation of MCL-1 [16]. Antileukemic synergistic effects of FLT3i and venetoclax are currently being evaluated in several clinical trials [17-19].
- Please clarify :” MRD analysis was performed on 317 of 368 patients that achieved CR after 1-2 courses of induction. CRc with MRD of <10-4 correlated with improved overall survival. While proportion of patients with MRD<104 was comparable between study and control groups, the rate of MRD negativity was significantly higher following treatment with quizartinib (13.8% vs 7.4%, p=0.017) [51].”
Paragraph has been rephrased as per reviewer’s suggestion. It currently reads as follows:
MRD analysis was performed in samples from 317 out of 368 patients that achieved CR after 1-2 courses of induction. CRc with MRD of <10-4 correlated with improved overall survival. While proportion of patients with MRD <10-4 was comparable between study and control groups (24.6% and 21.4%, p=0.385), the proportion of patients with unde-tectable MRD <10-5 using PCR-NGS technique was significantly higher following treatment with quizartinib (13.8% vs 7.4%, p=0.017), indicating improved depth of re-mission with use of FLT3i [55].
-After “CXCR4 expressed on leukemic cells and its ligand CXCR12 facilitate their homing to the bone marrow where they are nourished within the environment rich in growth-promoting and anti-apoptotic signals [59].”, what about the aforementioned drug metabolism ?
We would like to ask for clarification from the reviewer on whether they are referring to metabolism of FLT3i within the bone marrow microenvironment. If so, we have included this in our manuscript as follows:
FLTi are primarily metabolized by hepatic CYP3A4 and their bioavailability may vary with concurrent use of moderate and strong CYP3A4 inhibitors [64, 65]. Bone marrow stromal cells have been found to express CYP3A4 where it protects hematopoietic cells from toxic insults. Thus, local inactivation of FLT3i by CYP3A4 within bone marrow environment is yet another mechanism of resistance (Figure 1, pathway 4) [66].
- I suggest eliminating this period “Understanding the drivers for emergence of secondary resistance mechanisms are critical to overcoming them.”
The sentence has been eliminated as per reviewer’s suggestion.
- I think it might be interesting to add figures summarizing the mechanisms of resistance
We agree with the reviewer that visual representation of multiple resistance mechanisms would be beneficial. We generated a figure summarizing all of the resistance mechanisms we discussed in the text. Please see the figure within the manuscript file.

Round 2
Reviewer 1 Report
After they generated a figure to summarize mechanisms of resistance to FLT3i, it’s more friendly to readers.
There are still some spelling mistakes need to be corrected, such as the last sentence under the section “3. FLT3 inhibitors in relapsed refractory setting” , “additive 7myelosuppression” has an extra “7”.
Author Response
Dear Reviewer
Thank you again for your feedback. We are glad that you found the figure helpful. In addition to proofreading the manuscript ourselves we used a software program to check for spelling and grammar mistakes as per your suggestion.
Reviewer 2 Report
Thank you!
Author Response
Dear reviewer
Thank you for taking your time to read our manuscript and provide thoughtful feedback.